# Rare Earth Elements in the Seagrass *Zostera noltei* and Sediments from the Black Sea Coast of Crimea

**Vitaliy I. Ryabushko [1], Sergey V. Kapranov [1], Elena V. Gureeva [1], Nikolay I. Bobko [1] and Sophia S. Barinova [2,\*]**

[1] A.O. Kovalevsky Institute of Biology of the Southern Seas of RAS, 2 Nakhimov Av., 299011 Sevastopol, Russia; rabushko2006@yandex.ru (V.I.R.); arcanzas@rambler.ru (S.V.K.); lpopova09@rambler.ru (E.V.G.); ni.bobko@yandex.ru (N.I.B.)

[2] Institute of Evolution, University of Haifa, Mount Carmel, 199 Abba Khoushi Ave., Haifa 498838, Israel

\* Correspondence: sbarinova@univ.haifa.ac.il

**Abstract:** In the present work, we assessed the contents of rare earth elements (REEs), including yttrium and scandium, in rhizomes and leaves of the widespread seagrass *Zostera noltei* Hornemann 1832 and in the nearby sediments from the Black Sea coast. The total REE content in the sediments was found to be much higher than in *Z. noltei*. The order of decrease in the major REE contents in the sediments and the seagrass rhizomes was identical, except for La and Y. La was the most abundant in the sediments, and Y in the rhizomes. The contents of all REEs in rhizomes of *Z. noltei* were 1.5–10 times higher than in the leaves. The highest difference in the REE contents was found for the minor elements (Sm–Lu). The translocation factors for Sc and the minor elements (excluding Tb) from the sediments to the rhizomes and from the rhizomes to the leaves turned out to be pairwise equal, which indicates the similarity of the REE translocation mechanisms. Comparing our results with the literature data, it is possible to conclude that the seagrass *Z. noltei* does not have an advantage in the REE accumulation over marine macroalgae. At the same time, large coastal deposits of this seagrass after storms allow us to consider it as a possible source of REEs in the future.

**Keywords:** biomonitoring; leaves; rhizomes; translocation factor; bioremediation; beach wrack

## 1. Introduction

Environmental pollution by various types of toxic inorganic, organic, and organometallic compounds is one of the most serious environmental issues in the world. Rare earth element (REE) pollution in the marine environment is an emerging problem that has unpredictable consequences for marine ecosystems.

According to the definition by the International Union of Pure and Applied Chemistry (IUPAC), rare earth elements are a group of 17 elements, including 15 lanthanoids (metals from lanthanum to lutetium), scandium, and yttrium [1]. All rare earth elements are relatively abundant but scattered in the earth's crust, and they are found in more than 200 minerals, such as carbonates, silicates, and phosphates. The exception is the radioactive promethium, which is extremely rare in nature as it has no stable or long-lived isotopes [2]. Due to their unique physical and chemical properties, REEs are of decisive importance for the development of a multitude of industrial technologies and products from cell phones and TVs to LEDs and wind turbines. These elements are indispensable components of magnetic, optical, and electronic devices used in the defense and aerospace industries for the production of unmanned aerial vehicles, guided missiles, laser guidance devices for satellite communications, etc. [3–5]. Rare earth elements are used in agriculture as supplements to fertilizers to improve crop growth and production and, thus, are accumulated in the soil [6,7]. As the use of rare earth elements increases, their release into the biosphere will inevitably grow [8,9].

There are a large number of studies on adverse effects of notoriously toxic elements, such as As, Pb, Cd, and Hg [10,11] on biota. However, many other elements, including rare

earth elements, are increasingly used in industry, and the dumping of huge amounts of waste contributes to their ingress into the soil and groundwater. Contemporary studies show that REEs are present in almost all parts of the human body in low concentrations and may play an important biological role [12]. On the other hand, the REE concentration in drinking water exceeding the ordinary levels in potable water by a factor of seven may pose a potential risk to health [13]. The background REE levels in waters vary greatly and depend mainly on local geology. Under natural conditions, REEs are transferred only in small amounts through groundwater and the atmosphere, but their widespread use in recent times has increased the REE pollution and created new routes of REE bioaccumulation (in plants, animals, and humans). The use of modern analytical methods, such as inductively coupled plasma mass spectrometry (ICP-MS), helps improve our understanding of the reactivity and mobility of REEs in the near-surface environment, their bioavailability, and the possible risks to human health [14–16]. At the same time, the information on REE effects on human health is very limited at present [17,18].

The determination of REEs in marine aquatic organisms is an important aspect of studying the distribution of these elements in the environment. Currently, the REE levels in the flora and fauna of aquatic systems are not considered hazardous, but with the increasing use of these elements, the situation may change for the worse in the near future [19]. High degrees of REE bioaccumulation were found in phytoplankton [13] and in mussel shells in rivers of Europe and North America [20], proving the bioavailability of REEs and their ability to accumulate along the food chain. A possible risk to the environment lies in the bioavailability of REEs and their tendency to fractionate depending on environmental conditions, especially on the water pH and the nature of the solids [19]. Sediment characteristics such as Fe content can limit the transport of rare earth elements due to complexation reactions [21]. Currently, there is evidence of elevated levels of REEs in various natural systems, for example, of an anomalously high content of Ce mainly in the warm sea surface layer of the Black Sea near the Bosporus [22] and Gd in coastal waters near densely populated areas with developed medical care facilities due to the use of Gd as a contrast agent in magnetic resonance imaging [19].

The quantitation of REEs in marine biota is important for determining their levels in the environment. Along with other macrophytes, seagrasses are an important link in the matter and energy cycles in coastal ecosystems [23,24]. Therefore, seagrasses can serve as efficient bioindicators of REE pollution in the marine environment [25]. Moreover, they can be useful in the REE extraction from natural waters and wastewater, with the prospect of achieving REE recycling and water bioremediation [26,27].

In the world, the net primary production of seagrass meadows is about ~$0.6 \times 10^{15}$ g C per year [28]. Seagrasses perform numerous crucial ecological functions. They have the ability to modify sediment characteristics and provide stability to marine sediments. Additionally, they induce changes in the hydrodynamics of currents and waves. Furthermore, they act as natural barriers, limiting the proliferation of bacterial pathogens that pose a threat to human health. Moreover, seagrasses serve as critical habitats for various marine species, playing a fundamental role in the preservation of biodiversity [29].

Seagrass roots and rhizomes can be considered important natural archivers of anthropogenic activity due to their perennial and attached life in sediments. By accumulating and immobilizing contaminants from sediments, they can be used as indicators of long-term changes in metal fluxes in the coastal ecosystem. On the other hand, seagrasses shed leaves every year, and for this reason, seagrass leaves can serve as an indicator of the relatively short-term REE pollution in the marine environment [30–32]. Currently, there is very little information on interactions between seagrasses and pollutants, including REEs, and estimates of the REE transfer from sediments to seagrass [32]. The purpose of this work is to study the distribution of REEs in the rhizomes and leaves of the seagrass *Zostera noltei* and in the nearby sediments from the coastal area of the Black Sea. This information allows us to assess the usability of the seagrass as a possible indicator of REE pollution and as a potential agent for the recovery and bioremediation of REEs.

## 2. Materials and Methods

The sampling area was Kazachya Bay, the westernmost bay in Heracles Peninsula in the southwest of Crimea. Kazachya Bay is considered one of the cleanest bays in the area of Sevastopol [33,34], and this is one of the factors determining the high biodiversity in both the marine and adjacent terrestrial zones. The water temperature in the bay was reported to vary from 7.6 in February to 27.2 °C in July and August. The salinity variations are from 17.43 to 18.12‰ [33]. Sediments in Kazachya Bay are represented by silted sand with pebbles, silted shell sand, and, to a lesser extent, aleuritopelitic sand [35].

Five live specimens of the seagrass *Z. noltei* (=*Nanozostera noltei* (Hornemann) Tomlinson & Posluszny 2001, =*Z. noltii*) were randomly sampled in Kazachya Bay in June 2021 at one site within an area of about 5 m² at a depth of 0.5–1 m (Figure 1). The samples were placed in acid-rinsed polyethylene zip slider bags and within one hour delivered to the laboratory on ice. After removal of visible epiphytes, leaves and rhizomes of *Z. noltei* were rinsed with distilled water to further remove salts, detrital particles, and epiphytes. Approximately 20 g of fresh seagrass biomass (with leaves separated from rhizomes using a plastic knife) was dried to constant weight at 105 °C and cut into small pieces. For the REE quantitation, five samples of dry leaves and rhizomes (20 mg each) from each specimen were used, and the total weight of the dry biomass used was 200 mg (100 mg in leaves and 100 mg in rhizomes).

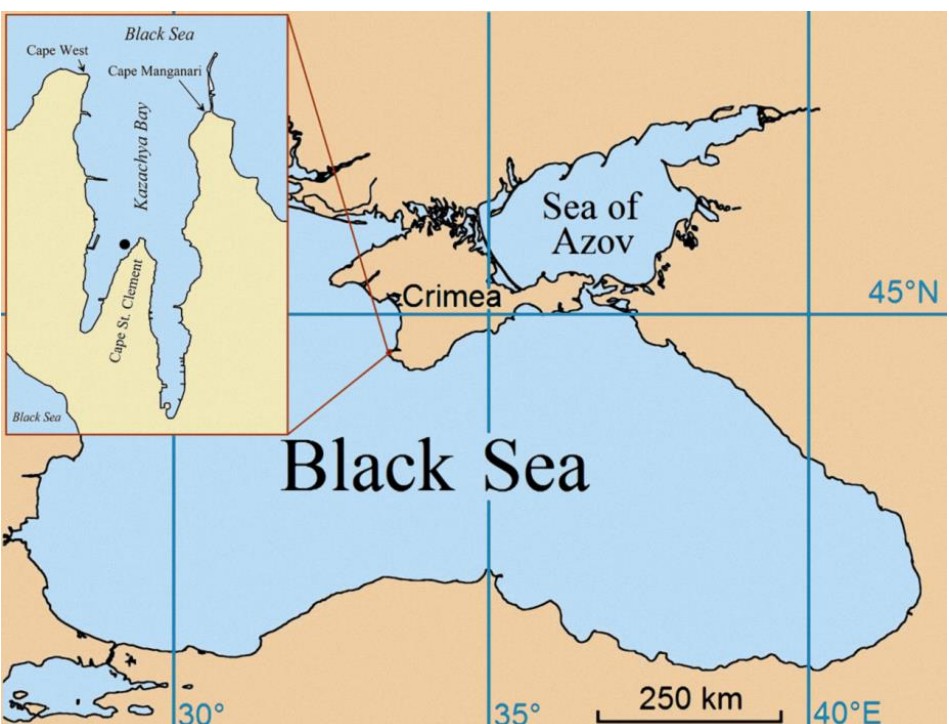

**Figure 1.** Map of the sampling area (inset). The sampling site is denoted by a black circle.

Sediments with a total weight of approximately 300 g were sampled manually from four different, randomly selected, nearby sites at a depth of 0.5–1 m using a plastic spatula. The thickness of the sampled sediment layer was 2–3 cm. The sediments were thoroughly mixed to eliminate local heterogeneities in the content of elements, dried to constant weight, ground in a porcelain mortar, and sifted through a sieve with a mesh size of 0.5 mm. Five sample replicates were taken for leaves, rhizomes, and sediments.

The sample digestion was performed for two hours using analytical-grade nitric acid purified by sub-boiling distillation in an acid purification system DST-1000 (Savillex, Eden Prairie, MN, USA) in capped PTFE tubes in an autoclave at 2.2 bar. The digested samples were diluted with deionized water to 1000 mL·g$^{-1}$ d.w. The REE and macroelement

concentration in the samples were measured on an ICP-MS instrument PlasmaQuant® MS Elite (Analytik Jena, Jena, Germany) in a single scan. The REE calibration curves were plotted using the blank solution and standard solutions with concentrations of 0.001, 0.005, 0.01, 0.05, 0.1, and 0.5 $\mu g \cdot L^{-1}$, prepared from multielement standards IV-ICPMS-71A,D (Inorganic Ventures, Christiansburg, VA, USA). For the macroelement calibration curves, the standard concentrations of 100, 300, and 500 $\mu g \cdot L^{-1}$ were used. In case the analyte concentration in a sample exceeded the upper calibration limit, the sample was diluted with deionized water so that the concentration fell within the calibration range. The coefficient of determination $R^2$ for all calibration curves was no smaller than 0.998.

The limits of REE detection on this instrument at this dilution are no higher than 0.03 $\mu g \cdot kg^{-1}$ d.w. [36]. Quality assurance and quality control of this analysis was provided via trace element quantitation in certified reference materials ERM®-CD200 (seaweed *Fucus vesiculosus*, Joint Research Centre, Institute for Reference Materials and Measurements, Geel, Belgium) and BCR-670 (duckweed *Lemna minor*, Joint Research Centre, Institute for Reference Materials and Measurements, Geel, Belgium) after digestion in nitric acid according to the above procedure.

Each analyte was measured in five replications, with five scans per replication. The plasma flow was 9.0 $L \cdot min^{-1}$, and the RF power was 1.25 kW. The dwell time for each analyte was 50 ms, with one point per peak in the peak-hopping mode. To ensure the insignificance of polyatomic interferences, collision reaction interface (CRI) was used as well. In the CRI, hydrogen with a flow rate 40 $mL \cdot min^{-1}$ was the skimmer gas. No internal standard was used. The signal drift was taken into account by measuring the element concentrations in the diluted standard IV-ICPMS-71A,D after every fifth sample.

Statistical analysis of the measurement results was carried out using one-way ANOVA in PAST 4.12 [37]. The post hoc pairwise comparison was performed using Tukey's test. In case of heterogeneity of variances detected in Levene's test, the averages were compared according to the Welch F-test and pairwise Games–Howell test in Matlab 8.2.0 [38]. Hierarchical cluster analysis was performed using PAST 4.12, with the Euclidean distance as the similarity measure and unweighted pair-group average (UPGMA) as the pairwise clustering algorithm.

To estimate whether or not REEs in the sediments originate mainly from natural sources, the enrichment factor (EF) [39,40] was calculated as follows:

$$EF = (REE/El)_s / (REE/El)_{ref} \tag{1}$$

where the subscript "s" denotes sediments and "ref" denotes some reference level unaffected by anthropogenic activity, and El is the content of elements (typically, macroelements) subject to a similar degree of weathering. Although the upper continental crust (UCC) values [41,42] are sometimes taken as reference [39,40] because the corresponding average values may represent the average source rock composition [43], enrichment factors for individual metals are highly dependent on the specific area under study [40]. For this reason, we adopt the REE levels in Late Pleistocene sediments from Sevastopol Bay [44] as reference values, taking into account that estimates from earlier epochs or continental crust for this area are not available.

The REE transfer from sediments to the different parts of the seagrass and the translocation from rhizomes to leaves are characterized with the transfer factors (TF) and translocation factor (TF'), respectively, which can be expressed in a straightforward manner as

$$TF = C_{seagrass} / C_s \tag{2}$$

where "seagrass" stands for either seagrass rhizomes or leaves, and

$$TF' = C_l / C_r \tag{3}$$

where the subscripts "l" and "r" denote leaves and rhizomes, respectively. TF > 1 indicates the REE bioaccumulation in a marine organism, and the greater the TF, the more mobile and bioaccessible the element forms are in the sediment environment [45].

## 3. Results and Discussion

### 3.1. REE Contents in the Seagrass and Sediments

The REE contents in the sediments were much higher than those in different parts of the seagrass (Table 1). The contents of the light REEs in *Z. noltei* rhizomes decreased in the following order: Y > La > Ce > Sc > Nd > Pr. For *Z. noltei* leaves, this order was different: Sc > La > Ce > Y > Pr > Nd. In the sediments, the Nd content exceeded that of Sc, and the sequence changed to La > Y > Ce > Nd > Sc > Pr. Thus, the elements in rhizomes and sediments, except for La and Y, had the same decreasing order. A similar order of the element decrease was observed in Bohai Bay situated west of the Bohai Sea, but the major element was Ce [46]. The coastal region surrounding Bohai Bay is one of China's most densely populated and industrialized areas, where approximately 97% of REE in the total Chinese output are produced, and industrial and agricultural use of the elements is growing rapidly [47]. Compared with this region, the concentration of REEs in the Black Sea, according to our data, was lower by a factor of 2 for La and 8–11 for all other REEs [46], which indicates a relatively low concentration of elements in the sediments of the Crimean coast of the Black Sea.

**Table 1.** REE contents (in $\mu g \cdot kg^{-1}$ d.w.) in leaves and rhizomes of *Zostera noltei* and in sediments: mean $\pm$ SD; transfer factors for rhizomes (TF$_r$) and leaves (TF$_l$); and rhizome–leaf translocation factors (TF′). The different upper subscript letters denote significant differences between the element contents in the objects of study: [a] < [b] < [c].

|  | *Z. noltei* (Leaves) | *Z. noltei* (Rhizomes) | Sediments | TF$_r$ | TF$_l$ | TF′ |
|---|---|---|---|---|---|---|
| Sc | 373 $\pm$ 22 [a] | 560 $\pm$ 15 [a] | 1382 $\pm$ 77 [b] | 0.405 | 0.270 | 0.666 |
| Y | 153 $\pm$ 9 [a] | 1562 $\pm$ 18 [b] | 12,221 $\pm$ 646 [c] | 0.128 | 0.012 | 0.098 |
| La | 243 $\pm$ 26 [a] | 761 $\pm$ 112 [a] | 17,074 $\pm$ 614 [b] | 0.045 | 0.014 | 0.320 |
| Ce | 221 $\pm$ 9 [a] | 648 $\pm$ 25 [b] | 6232 $\pm$ 89 [c] | 0.104 | 0.035 | 0.341 |
| Pr | 87 $\pm$ 3 [a] | 174 $\pm$ 11 [b] | 1162 $\pm$ 18 [c] | 0.150 | 0.075 | 0.499 |
| Nd | 38 $\pm$ 3 [a] | 267 $\pm$ 49 [b] | 2894 $\pm$ 37 [c] | 0.092 | 0.013 | 0.145 |
| Sm | 8.2 $\pm$ 0.4 [a] | 65 $\pm$ 6 [b] | 635 $\pm$ 19 [c] | 0.102 | 0.013 | 0.126 |
| Eu | 2.6 $\pm$ 0.4 [a] | 21.7 $\pm$ 1.3 [b] | 175 $\pm$ 6 [c] | 0.124 | 0.015 | 0.122 |
| Gd | 10.4 $\pm$ 0.3 [a] | 75 $\pm$ 6 [b] | 671 $\pm$ 16 [c] | 0.112 | 0.015 | 0.138 |
| Tb | 6 $\pm$ 1 [a] | 22 $\pm$ 3 [b] | 184 $\pm$ 8 [c] | 0.121 | 0.033 | 0.270 |
| Dy | 11 $\pm$ 1 [a] | 72 $\pm$ 7 [b] | 551 $\pm$ 27 [c] | 0.131 | 0.019 | 0.146 |
| Ho | 2.4 $\pm$ 0.3 [a] | 19 $\pm$ 1 [b] | 131 $\pm$ 4 [c] | 0.144 | 0.018 | 0.125 |
| Er | 7.3 $\pm$ 0.6 [a] | 53 $\pm$ 4 [b] | 386 $\pm$ 13 [c] | 0.136 | 0.019 | 0.139 |
| Tm | 1.5 $\pm$ 0.1 [a] | 9 $\pm$ 2 [b] | 49 $\pm$ 2 [c] | 0.190 | 0.031 | 0.164 |
| Yb | 5.1 $\pm$ 0.7 [a] | 35 $\pm$ 2 [b] | 241 $\pm$ 13 [c] | 0.145 | 0.021 | 0.146 |
| Lu | 1.7 $\pm$ 0.2 [a] | 9.3 $\pm$ 3.4 [b] | 45 $\pm$ 2 [c] | 0.206 | 0.038 | 0.183 |
| ΣREE | 1171.2 | 4353 | 44,033 |  |  |  |

On the whole, the abundance of all REEs in rhizomes was 1.5–10 times higher than in leaves. Moreover, the contents of heavy REEs (Sm–Lu) differed by a factor of 7–8 (Table 1). Among all REEs, the highest content was noted for Y in the rhizomes of *Z. noltei*. In the sediments, the highest content was registered for La. In general, the order of the REEs decrease in different objects was as follows: sediments > rhizomes > leaves.

The cluster analysis of the REE contents in the seagrass (Figure 2) showed the division of elements into two groups: major (Sc, Y, La–Nd) and minor (Sm–Lu) ones. Major REEs are characterized by the levels of hundreds to thousands $\mu g \cdot kg^{-1}$ in rhizomes and tens to hundreds $\mu g \cdot kg^{-1}$ in leaves. Minor REEs have the values of tens to hundreds $\mu g \cdot kg^{-1}$ in rhizomes and up to ten $\mu g \cdot kg^{-1}$ in leaves. The abundance of REEs in marine organisms,

excluding filter feeders, is due to the element contents in sediments and suspended particles rather than to the amount of REEs dissolved in seawater [48]. In seawater, light REEs are to a greater extent absorbed by organisms or adsorbed on their surfaces, and the small ionic radii of heavy REEs allow them to remain in the solution due to the formation of stable complexes [19]. There is a small percentage of free ionic REEs, mainly light REEs, in the solution, which are easily incorporated into organisms [19,49] and may potentially have serious biological consequences. Another explanation for the fractionation of REEs in seawater is associated with diatoms [50], which absorb these elements in the sea surface layer and transport them to the deeper waters. Light REEs tend to be bound to carbonates and oxides and are thus more accumulated than heavy REEs in the surficial layer of sediments. This may serve as an explanation for the enrichment of ocean waters with heavy REEs.

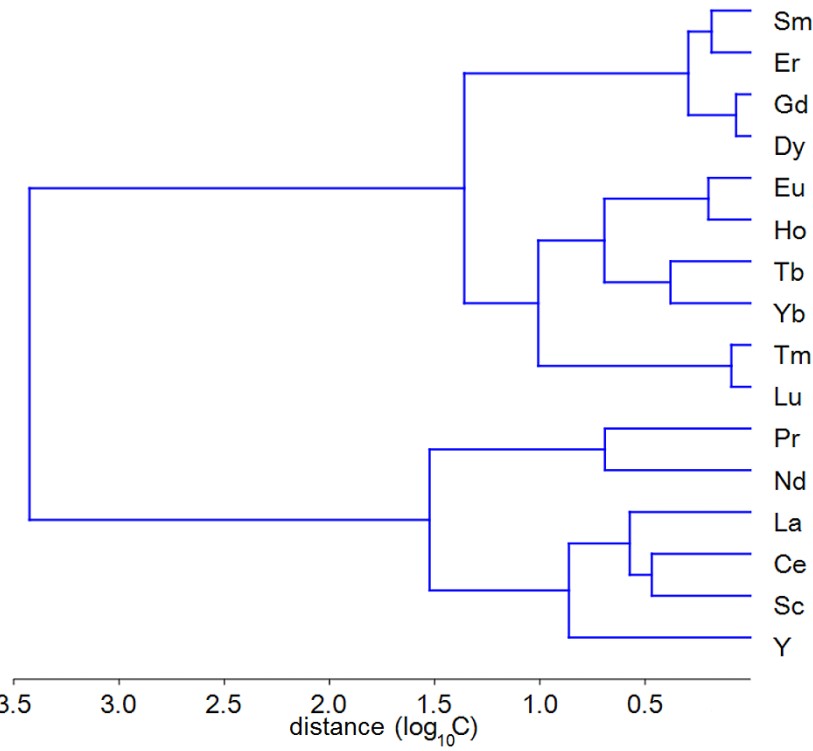

**Figure 2.** Dendrogram of the Euclidean distances of log-transformed REE contents in rhizomes and leaves of *Zostera noltei*.

There is a lack of data on the content of REEs in terrestrial and marine organisms, especially in the latter. Therefore, the role of REEs in marine organisms requires further research. One of a few reviews on REEs in aquatic biota [19] summarized data on REEs in various marine organisms, such as algae, mollusks, crustaceans, corals, and fish. Low REE levels were found in wild corals, and even lower ones were found in fish. The highest content of these elements was found in algae; in addition, algae can accurately indicate the provenance of REEs, seawater, or sediments [51]. For example, the macroalga *Gracilaria gracilis* can efficiently absorb and trap REEs from wastewater with low REE concentrations in it [52]. The REE accumulation in marine organisms may depend on trophic level; however, biomagnification does not appear to occur [19].

The total REE content in the seagrass *Z. noltei* from the Black Sea is 4–20 times lower than in the seagrass *Halodule wrightii* collected in Todos os Santos Bay on the east coast of Brazil [32]. It is noteworthy that the total REE content in sediments on the coast of Brazil was only 1.2–2 times higher than in the plant. This indicates that the ability of *Z. noltei* to accumulate REEs is relatively low. Unlike *Z. noltei*, no significant difference in the REE contents was registered between the roots and leaves of *H. wrightii*. The authors of [32] also

concluded that seagrasses and their tissues have a low potential for REE bioaccumulation, as the REE levels in sediments did not correlate with those in the plant.

In another seagrass species, *Cymodocea nodosa*, the REE contents were significantly higher in the shoots and not in the underground parts of the plant, with the light REEs (La, Ce, Pr, Nd) being 1.7 times higher in abundance in above-ground parts than in roots and rhizomes [39]. The contents of La–Lu in *C. nodosa* leaves were an order of magnitude higher than those obtained in the present study. In the underground parts, the REE contents were comparable to our data except for Ce, Nd, Sm, and Gd, which were also higher in *C. nodosa*. The contents of Pr–Lu, except for Tb, in sediments were 5.5–7 times higher, and the contents of Ce and La were 9 and 1.5 times higher, respectively. It is noteworthy that the REEs in the rhizomes of *Z. noltei* were almost equal to those in *C. nodosa*, with the abundance in sediments being significantly lower.

The difference in the REE contents in seagrasses can be explained, among other possible reasons, by different physicochemical properties of the sediments. The REE absorption through roots is affected by the concentration of iron oxyhydroxide, pH, and redox potential in the rhizosphere and the cation exchange capacity of sediments. Organic and inorganic ligands also play an important role in the REE uptake by plant roots and REE speciation, and affinity for plasma membrane affects the uptake of individual elements [53]. The REE distributions in plants and in sediments are typically dissimilar, and the contents of REEs in plants do not correlate with those in the rhizosphere [53], probably due to the plant's control of REE uptake [54].

The biochemical functions of trace elements are extremely complex, as they involve interactions with other elements and can have dose-dependent effects of different signs and intensities [55]. To date, the biological role of REEs remains unknown, but it was shown in some studies that they can have both a negative and positive influence on animals [56,57]. As a result, it has been shown that REEs are involved in the activity of a wide range of enzymes; it was also discovered that rare earth elements, namely $Ce^{3+}$ and $La^{3+}$ ions, are necessary for some bacteria to promote methanol oxidation reaction [58]. The biochemical action of rare earth elements is largely due to the closeness of their ionic radii to those of other, essential elements. For example, the $Ca^{2+}$ cation can be replaced by trivalent lanthanide ions at calcium binding sites in biological molecules [57]. The largest amount of $Ca^{2+}$ is contained in the plant cell wall in the form of calcium pectinate. By replacing $Ca^{2+}$, REEs form complexes with pectin, and their trivalency gives them a much higher charge-to-volume ratio, which means that they have a much higher affinity than $Ca^{2+}$ for these binding sites [59,60]. REE are typically assimilated to a greater extent than other non-essential elements. For example, $La^{3+}$ and $Eu^{3+}$ were found in the membranes of chloroplasts, mitochondria, cytoplasm, and nuclei [53,61]. REEs are evenly distributed in the membranes of chloroplasts and thylakoids, where these elements are associated with the photosystem II complex [60].

### 3.2. REE Enrichment and Anomalies in Sediments

To calculate the enrichment factors (EF) of REEs in sediments according to Equation (1), they were normalized to the content of Mg (0.256%) and P (0.010%), since they were relatively constant throughout the Holocene and close to the values in the Upper Pleistocene sediments (0.329% and 0.024%, respectively) [46]. It is seen that the enrichment factors of Sc, Y, La, and Tb exceed 1 (Figure 3) and are likely related to anthropogenic activity. The enrichment in these elements may result from landfill erosion and/or leaching of phosphate fertilizers containing REE additives.

The enrichment factors of the other REEs are below or close to 1, which indicates the natural provenance of these elements. The latter case (EF ≈ 1) is seen for the heaviest REE (Ho–Lu) when normalized to the phosphorus content. The REE depletion can be attributed to the dominance of coarse-grained fractions of sand and carbonate minerals in the sediments, as fine-grained fractions of silt and clay are typically enriched in REEs [39]. There is a trend for the REE enrichment factor to increase with the atomic weight of the

element. Such enrichment in heavy REEs was observed in the coarse-grained fraction of river sediments from South China due to the higher percentage of heavy minerals, such as zircon, monazite, and amphibole [62]. The enrichment in heavy REEs can also result from multiple cycles of element adsorption from the water column by suspended organics (including phytoplankton) and its subsequent sedimentation, resuspension, and remobilization [63].

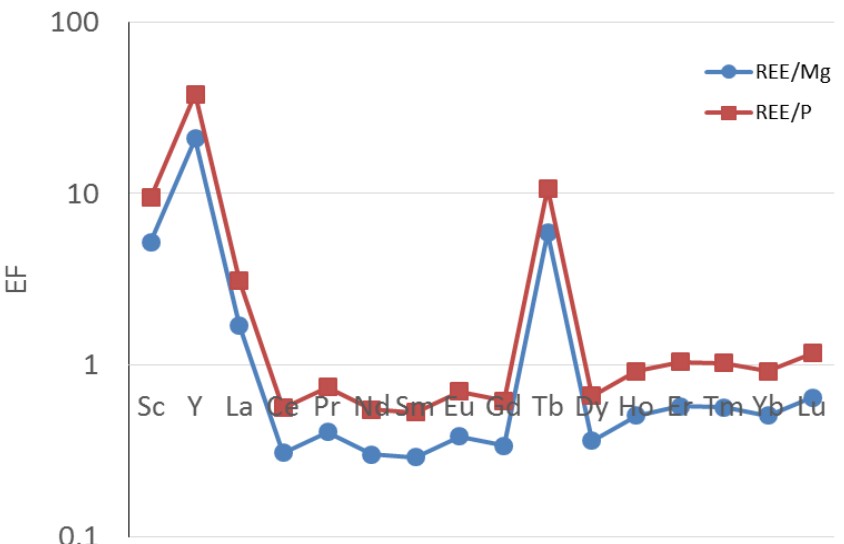

**Figure 3.** Enrichment factors of REEs (1) in the sediments from Kazachya Bay.

Anomalies of Ce and Eu are important indicators of biogeochemical conditions in the environment [39,43]. Their most characteristic implication is the information on the surrounding redox conditions. A relative depletion in Ce, when insoluble $CeO_2$ is reduced to soluble $Ce^{3+}$, represents an example of the negative Ce anomaly and indicates the reductive conditions, whereas the relative Ce enrichment is the positive Ce anomaly that suggests an oxidative environment. The Eu anomalies are opposite to those of Ce in terms of reaction to redox conditions and are typically considered in mineralogy rather than in life sciences. In the reductive conditions, ions of $Eu^{2+}$ substitute for $Ca^{2+}$ in some minerals such as feldspars in hydrothermal, metamorphic, and magmatic processes and form a positive Eu anomaly, whereas in oxidative conditions, Eu occurs in the form of $Eu^{3+}$ ions, which do not have such an ability, and a negative or no Eu anomaly is observed. The element anomalies can be calculated using the following relationships [39]:

$$\delta Ce = Ce_N/(La_N \times Pr_N)^{1/2} \tag{4}$$

and

$$\delta Eu = Eu_N/(Sm_N \times Gd_N)^{1/2} \tag{5}$$

where $Ce_N$, $La_N$, $Pr_N$, $Eu_N$, $Sm_N$, and $Gd_N$ are the mean values normalized to the corresponding Late Pleistocene values from a sediment core at a nearby site [44].

In our case, $\delta Eu = 1.16$ did not essentially differ from the values for the other REEs calculated according to similar relationships (0.73–1.57), which implies the absence of any Eu anomaly. However, $\delta Ce = 0.33$ was substantially lower, and this suggests the existence of a negative Ce anomaly in the sediments. The negative Ce anomaly indicates a reductive environment in sediments of Kazachya Bay, which is due to the decay of organic material settling to the seafloor. A similar anomaly was detected in other sediments in reductive environments [39,64].

### 3.3. REE Transfer and Translocation in the Seagrass

Transfer factors associated with the transfer from sediments to rhizomes and leaves ($TF_r$ and $TF_l$) and the rhizome–leaf translocation factors ($TF'$) are presented in Table 1. All the factors are below 1, which confirms the low efficiency of REE uptake by the plant. The sediment–rhizome transfer factors are up to an order of magnitude higher than the sediment–leaf transfer factors, with the efficiency of the REE transfer to leaves decreasing in the following order: Sc >> Pr > Lu > Ce > Tb > Tm > Yb > Dy = Er > Ho > Eu = Gd > La > Nd = Sm > Y. This fact is at odds with the data of the work [39], where a higher REE transfer to the leaves of the seagrass *Cymodocea nodosa* was detected.

In Figure 4, the ratio of the REE contents in rhizomes and leaves ($1/TF'$) and the ratio of the REE contents in the sediments and rhizomes ($1/TF_r$) are shown. For Sc and minor REEs, except for Tb, these ratios turned out to be almost pairwise equal, which implies the similarity of the mechanisms of their accumulation and translocation. To explain this phenomenon, it can be assumed that the differences in the REE content ($\Delta C$) between sediments and rhizomes and rhizomes and leaves are proportional to their content (C) in different parts of the plant and differences in some unspecified parameter q ($\Delta q$) that are assumed equal for the sediment–rhizome and rhizome–leaf "interfaces":

$$\Delta C = kC\,\Delta q \tag{6}$$

where k is the proportionality factor that shows the steepness of the concentration increment with the parameter q increase.

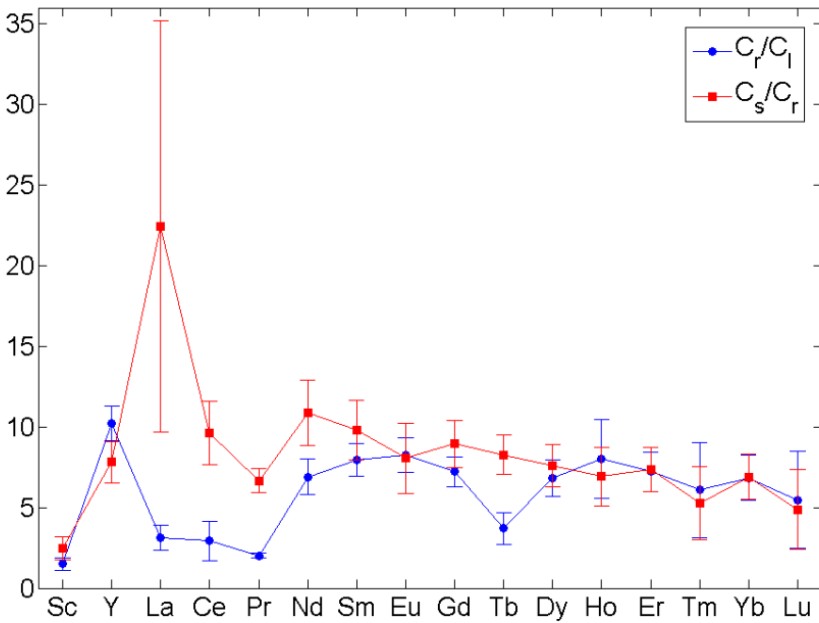

**Figure 4.** Ratios between the REE contents in sediments and rhizomes ($C_s/C_r$) and in rhizomes and leaves ($C_r/C_l$) of *Zostera noltei.* The whiskers represent 95% confidence intervals.

In the small differences approximation, one can look to the differential form of Equation (4):

$$dC = kC\,dq. \tag{7}$$

From Equation (7) it follows that

$$C = C_s \exp(kq) \tag{8}$$

where $C_s$ is the content in the sediments, and it is seen that equal intervals of q will change the content by an equal factor.

It is likely that the q variable is related to the local pH value (more precisely, to the difference between the local pH and the pH of pore water), which decreases in the direction from pore water to the xylem sap of the plant leaves [65]. It is well known that a pH decrease increases the solubility of sparingly soluble salts of REEa [53,54,66,67] and, consequently, reduces their ability to accumulate. Rare earth elements can be accumulated in seagrass tissues by binding to plant matrix carboxyl or phosphate groups nearly as tightly as to carbonates and phosphates in sediments; alternatively, REEs can deposit on the corresponding inorganic microgranules in tissues. This assumption finds indirect confirmation in the high correlation coefficients with calcium (median $r = 0.99$), which was also noted in the algae of the genus *Cystoseira* [68].

The changes in the $C_s/C_r$ ratios (Figure 3) represent the well-known tendency towards an increase in the solubility of lanthanide phosphates and carbonates with an atomic number increase [69,70]. The reduced values of $C_r/C_l$ for La–Nd and Tb can be explained by the substitution for yttrium in rhizomes and/or the formation of specific soluble complexes of these elements in leaves. On the whole, the seagrass *Z. noltei* does not have an advantage in the accumulation of REEs compared with other native macrophytes (*Cystoseira* spp.) [68] due to the low translocation efficiency through the root system, as was also noted for the South American seagrass *Halodule wrightii* [32].

*3.4. Seagrass Wrack as a Potential Source of REEs*

The availability and extractability of REEs from various natural sources other than minerals is an important issue associated with the commercial use of REEs in the future [71]. Currently, the ever-increasing number of applications of REEs and high added value of REE-containing products has led to an increase in REE production volumes. Therefore, research on minimizing industrial production losses and more cost-effective extraction of REEs with the least environmentally unfriendly consequences is in an increasing demand [72]. As a result of processing algal and seagrass biomass, including storm-cast deposits (beach wrack), it is possible to produce various types of biofuels, including biodiesel, bioethanol, biogas, biohydrogen, and other valuable products [73,74]. As storm-cast beach wrack of algae and seagrass can also be a source of REEs and bioremediate industrial wastewater, further research in this direction is strongly needed.

The Black Sea has extensive natural resources of macrophytes, including seagrasses. In late summer and early autumn, seagrasses shed their leaves, and after each storm, their mass deposits appear on the coastline (Figure 5). They are most widespread in the Kerch Strait and in Tendra, Dzharylgach, Yegorlyk, and Karkinit bays.

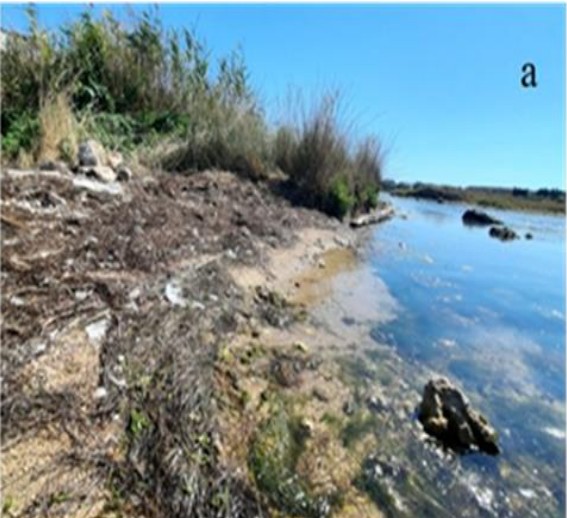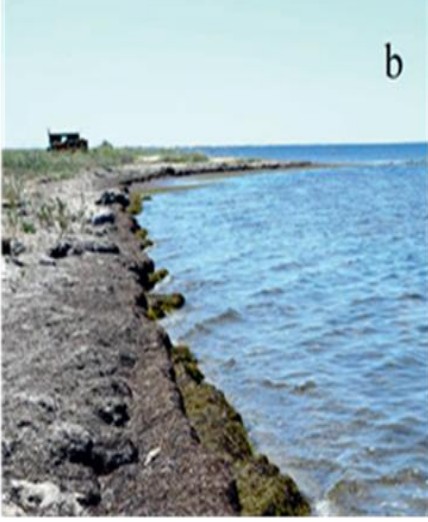

**Figure 5.** Seagrass deposits on the Black Sea coast after storm in (**a**) Kazachya Bay and (**b**) Karkinit Bay near the Portovoye settlement.

An assessment of storm-cast deposits of the seagrass as a possible source of REEs can be made. According to our data, the average REE concentration in the seagrass leaves is 0.4 mg·kg$^{-1}$ wet weight. After only one storm, the seagrass biomass washed ashore can be estimated at 10 tons per linear kilometer of coast in the southern part of Karkinit Bay in the northern Black Sea [75]. This biomass contains approximately 4 kg of rare earth elements. Estimating the reserves of the deposits of the seagrass on the Black Sea shoreline at hundreds of thousands of tons [76], one can arrive at REE reserves in the coastal seagrass deposits of up to 100 tons. However, a comprehensive environmental analysis is clearly needed for the large-scale use of the deposits of macrophytes for the extraction of REEs and other useful materials from them.

## 4. Conclusions

This work presents the first quantitative determination of a complete set of rare earth elements (except for promethium) in the seagrass *Z. noltei* and in the nearby bottom sediments. The significantly lower REE contents in *Z. noltei* rhizomes compared with their contents in the sediments have demonstrated the low ability of the seagrass to assimilate these elements, which may also be related to the specific characteristics of marine sediments in this area. Ratios of the contents of Sc and minor REEs, except for Tb, in the rhizomes and leaves of *Z. noltei* and in sediments and rhizomes have turned out to be very close. This fact probably indicates the similarity of mechanisms of the REE accumulation and translocation and is possibly associated with the pH decrease from the rhizosphere through the rhizomes to the leaves. These results can be useful for the further study of REEs in the Black Sea ecosystems and to monitor these elements in marine environments.

The mean REE content in *Z. noltei* leaves that are shed every year and are deposited as beach wracks has been estimated at 0.4 mg·kg$^{-1}$ wet weight. By utilizing hundreds of thousands of tons of *Z. noltei* wrack annually cast on the beaches of Crimea, up to 100 tons of REE can be yielded from this biomass. However, the environmental safety and cost-effectiveness of this technology requires further comprehensive research.

**Author Contributions:** Conceptualization, V.I.R.; Data curation, E.V.G. and S.V.K.; Methodology, S.V.K. and N.I.B.; Validation, S.V.K.; Writing—original draft, S.V.K. and E.V.G.; Writing—review and editing, V.I.R. and S.S.B. All authors have read and agreed to the published version of the manuscript.

**Funding:** This work was supported by the Russian Science Foundation grant #23-24-00494 (https://rscf.ru/project/23-24-00494/, accessed on 16 October 2023).

**Institutional Review Board Statement:** Not applicable.

**Informed Consent Statement:** Not applicable.

**Data Availability Statement:** The data that support the findings of this study are available from the corresponding author upon reasonable request.

**Acknowledgments:** We gratefully acknowledge the service of the Spectrometry and Chromatography core facility within IBSS RAS for the ICP-MS analysis and are thankful to A.M. Toichkin for his help in collecting seagrass samples.

**Conflicts of Interest:** The authors declare that they have no known competing financial interest or personal relationships that could have appeared to influence the work reported in this paper.

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
