# Peer review of "Rare Earth Elements in the Seagrass Zostera noltei and Sediments from the Black Sea Coast of Crimea"

_jmse, doi:10.3390/jmse11102021_

Round 1
Reviewer 1 Report
please see the annotated manuscript and the comments below.
Rare earth elements studies on marine ecosystems and sea grasses are not very limited but useful to understanding the behavior of different sea grasses species in different coastal ecosystems. In this study REE content in one sea-grass species found in the coastal area around Kazachya Bay, Crimea was assessed, and the possibility of using the particular grass species as a pollution indicator was determined.
The study itself proves that the REE accumulation in this particular seagrass is low and average and hence, I do not think that this plant can be used as a good indicator of REE pollution. Some sections in the manuscript are difficult to follow and do not make sense with what the authors trying to say. Another main point is the sampling location or sites were from the same area or same ecosystem. So I wonder whether would this make any sense if you compare only those results with UCC data. The study lacks sufficient data and only depends on the REE content in the grass species and the sediments.
Introduction –OK but would be good if more information is provided, e.g. importance of sea-grass meadows. Also, the information related to the accumulation of REE in different parts of the plants is missing. The research problem is not very clear.
Materials and Methods
Lacks important information such as sample preparation and the number of sites where sea grass was collected. For example, it is important to mention how different parts of the seagrass (i.e. leaves, rhizome) separated for the drying process.
Results and Discussion
Also, only the REE contents were measured, other physical and chemical parameters in sediments and sea can influence the REE content.
Why are the REE in seawater in the area not measured?
Section 3.1-Line 217-225-Better suits for Section 3.3
Section 3.1-Line 236-254-Better suits for the Introduction.
Section 3.3- The content in the 2nd paragraph and below are not very clear
Section 3.4- Does not contain any useful information.

Author Response
Dear Editor and the Reviewer 1,
We thank the Reviewer 1 for the valuable comments and suggestions that have helped us to considerably improve the quality of the manuscript. We have tried to take all comments into account and make all the necessary changes in the manuscript.
With best regards,
Prof Sophia Barinova,
Corresponding author
Responses to Reviewer 1 comments
Comment: Some sections in the manuscript are difficult to follow and do not make sense with what the authors trying to say
Reply: The manuscript has been essentially revised with the Reviewer’s general and specific comments taken into account. Many sections (e.g., 1, 2 and 3.2) have been rewritten or supplemented with additional information, and the Conclusions section has been completely rewritten. Hopefully, these modifications facilitate understanding of the ideas we are trying to convey. The object of research was seagrass since information on REE in seagrasses is extremely scarce and is limited to a couple of studies. Considerable attention has been paid to the apparent transfer and translocation factors and provenance of REE in sediments as the environment from which the seagrass extracts elements. Other factors of the REE accumulation were beyond the scope of this study.
Comment: Another main point is sampling location or sites were from the same area or same ecosystem. So I wonder whether would this make any sense if you compare only those results with UCC data.
Reply: We thank the Reviewer 1 for this comment. Indeed, the primordial background levels may differ in different areas. As the information on REE and reference elements in the average local rock composition is not available to us, we have used the data from local Late Pleistocene sediments. This has been emphasized in Materials and Methods in the revision.
Comment: The study lacks sufficient data and only depends on the REE content in the grass species and the sediments.
Reply: As mentioned above, we also used the data on REE in the local Late Pleistocene sediments (Merenkova et al., 2023) in the revision. The focus of this manuscript has been only on the seagrass and surrounding sediments as we were basically interested in studying REE in a representative of marine higher plants, which are extremely poorly studied to date in this respect. We have obtained data on the element contents in sediments, rhizomes and leaves of Z. noltei, observed a relationship between them (Cs/Cr=Cr/Cl) and proposed an analytical explanation to this relationship.
Comment Introduction –OK but would be good if more information is provided, e.g. importance of sea-grass meadows. Also, the information related to the accumulation of REE in different parts of the plants is missing. The research problem is not very clear.
Reply: The information on importance of seagrass meadows according to the Reviewer’s suggestion has been added in Introduction: “In the world, the net primary production of seagrass meadows is about ~0.6·1015 g C per year. Seagrasses fulfill numerous crucial ecological functions. They have the ability to modify sediment characteristics and provide stability to marine sediments. Additionally, they induce changes in the hydrodynamics of currents and waves and they act as natural barriers, limiting the proliferation of bacterial pathogens that pose a threat to human health. Moreover, seagrasses serve as critical habitats for various marine species, playing a fundamental role in the preservation of biodiversity.”
We studied REE contents in average samples of leaves of Z. noltei, in particular, to check the possibility of using it as a monitor since it is difficult to divide plants into parts when biomonitoring REE concentrations in the environment. In view of the lack of data on REE in the seagrass from the Black Sea, the research problems are the distribution of REE in sediments, rhizomes and leaves of Z. noltei and usability of Z. noltei as a potential biomonitor, bioremediator and source of REE.
Comment: Lacks important information such as sample preparation and the number of sites where sea grass was collected. For example, it is important to mention how different parts of the seagrass (i.e. leaves, rhizome) separated for the drying process.
Reply: We have added this information to Materials and Methods:
The sampling area was Kazachya Bay, the westernmost bay in Heracles Peninsula in the southwest of Crimea. Kazachya Bay is considered one of the cleanest bays in the area of Sevastopol (Kuftarkova et al., 2008; Soloveva et al., 2021), and this is one of the factors determining the great biodiversity in both the marine and adjacent terrestrial zones. Water temperature in the bay was reported to vary from 7.6 in February to 27.2 °C in July and August. The salinity variations are from 17.43 to 18.12 ‰ (Kuftarkova et al., 2008). Sediments in Kazachya Bay are represented by silted sand with pebbles, silted shell sand and, to a lesser extent, aleuritopelitic sand (Ignat'yeva et al., 2005).
Five live specimens of the seagrass Z. noltei were randomly sampled in Kazachya Bay in June 2021 at one site with the area of about 5 m2 at a depth of 0.5–1 m (Figure 1). The samples were placed in acid-rinsed polyethylene zip slider bags and within one hour delivered to the laboratory on ice. After removal of visible epiphytes, leaves and rhizomes of Z. noltei were rinsed with distilled water to further remove salts, detrital particles and epiphytes. Approximately 20 g of fresh seagrass biomass (with leaves separated from rhizomes using a plastic knife) were dried to constant weight at 105 °C and cut into small pieces. For the REE quantitation, five samples of dry leaves and rhizomes (20 mg each) from each specimen were used, and the total weight of the dry biomass used was 200 mg (100 mg in leaves and 100 mg in rhizomes).
Sediments with the total mass of approximately 300 g were sampled manually from four different, randomly selected, nearby sites at a depth of 0.5–1 m using a plastic spatula. The thickness of the sampled sediment layer was 2-3 cm. The sediments were thoroughly mixed to eliminate local heterogeneities in the content of elements, dried to constant weight, ground in a porcelain mortar, and sifted through a sieve with a mesh size of 0.5 mm. Five sample replicates were taken for leaves, rhizomes and sediments.
Comment: Also, only the REE contents were measured, other physical and chemical parameters in sediments and sea can influence the REE content.
Reply: Yes, a multitude of different physical and chemical parameters can influence the REE content. However, determining parameters that affect the REE accumulation was not the goal of our study. Moreover, due to the extreme complexity of live organisms, these parameters can affect it on different time scales and with different time lags, so it can be prohibitively difficult to conduct such a research. Our study was descriptive rather than hypothesis-driven.
Comment: Why are the REE in seawater in the area not measured?
Reply: In sediments, the temporal and spatial variation of trace metal concentrations is far less pronounced than in seawater. Furthermore, the REE contents in marine plants (Cystoseira spp.) demonstrate stronger correlations with those in sediments than in seawater (Ryabushko et al., 2022). Therefore, monitoring trace metal concentrations in sediments provides more meaningful and useful data for evaluating metal contamination in the coastal environment and predicting the effects trace metals may have on marine ecosystem (Hwang, 2016).
Comment: Section 3.1-Line 217-225-Better suits for Section 3.3
Reply: It seems to us that this paragraph is more relevant to the section with a general description of REE in seagrasses. It does not contain any information on element ratios in different parts of seagrasses, whereas Section 3.3 contains data on ratios between the REE contents in sediments and rhizomes (Cs/Cr) and in rhizomes and leaves (Cr/Cl).
Comment: Section 3.1-Line 236-254-Better suits for the Introduction.
Reply: We have removed the first sentence, which is too introductory. The rest of the paragraph, in our opinion, better suits for Discussion. In introduction, the literature that is relevant to the goalsetting is typically overviewed, while in discussion, different aspects and facets relevant to results and comparison with the literature data are presented. In our case, the paragraph contains information about biochemical role of REE in plants, which is more relevant to the contents of this section (REE Contents in the Seagrass and Sediments) rather than to the goal of the study and Introduction.
Comment: Section 3.3- The content in the 2nd paragraph and below are not very clear
Reply: We have introduced some modifications to this section to make it clearer. In particular, some of sentences have been rewritten. To explain the equality of the ratios Cs/Cr and Cr/Cl, the concentration (C) of specific REE along the plant xylem was introduced as a function of some variable q, which is assumed to have equal differences (Δq) at the “interfaces” between sediments and rhizome and between rhizome and leaf. We further assume that the concentration change at these “interfaces” is proportional to the local concentration value and Δq (Eq. (4)). Passing from finite increments (ΔC and Δq) to infinitesimal differentials and solving the differential equation (5), we obtain relationship (6) that explains the equality of the above ratios. Furthermore, we suppose that the parameter q is in fact the difference between local pH and pH of pore waters (i.d., rhizosphere). The rationale for this supposition is given in the next two paragraphs of this section.
Comment: Section 3.4- Does not contain any useful information.
Reply: We believe that possible sources of REE, in particular, from the storm-cast seagrass wrack, which is often considered a seashore pollutant and waste material, may be successfully utilized in the near future, given the high demand for REE in industry and agriculture. Therefore, in accordance with the manuscript purpose, we carried out a quantitative estimate of REE reserves in this material.
Replies to specific comments in the manuscript
L.18 Comment: Is the senetnces finish here, if so please put a period. This indicates the ratio of Y:La.
Reply: The sentence has been corrected according to the suggestion.
L.21 “minor” In this sentence we have decided to remove the classification of heavy and light REE since our division is not in full agreement with it. In particular, the light REE samarium and europium fall into the group of heavy REE. We have decided to leave only the division into major (Sc,Y, La, Ce, Pr, Nd) and minor REE, in accordance with the cluster analysis in Figure 2.
Reply: corrected
L.27 Comment: use words not in the Title
Reply: The keywords have been corrected as suggested.
L.32 Comment: (REE)
Reply: Accepted
L.33 Comment: Pure
Reply: Corrected
L.46 Comment: Rare earth elements
Reply: Accepted
L.72-73 Comment: surface layer of what? Black Sea water?
Reply: We have specified in revised ms that this is sea surface layer of the Black Sea near the Bosporus
L.73 Comment: in soils??
Reply: In coastal waters, added.
L.84 “to” – replaced by “into”
Reply: done
L.90-95 Comments:
not clear whether the sea grass were collected from only one place ? or from several places in the coast and if so what was the selection criteria for the selection of sampling sites?
Here, you need to state how you prepared a separate sets of samples of sea grass as a whole, rhizome of sea grass and leaves of sea grass.
depth?
Reply: Five specimens of the seagrass were sampled from a site with an area of 5 m2. Five samples of leaves and rhizomes were prepared from the five individual specimens after drying at 105 °C and crushing. The depth of the sediment sampling from the sea surface was 0.5–1 m, and the thickness of the sediment sample was 2-3 cm. All this information has been included in the Materials and Methods section, which has been essentially expanded.
L.153 Comment: This percentage is for china?
Reply: This percentage is for the total production in China. The information has been specified in the revision.
L.172 Comment: and
Reply: Corrected accordingly
L.196 Comment: ok, in where?
Reply: “REE concentration in drinking water exceeding the ordinary levels in potable water”. Moved to Introduction
L.215-216 “…as the highest REE values in sediments at one of the stations did not correspond to the highest levels of REE accumulation in the plants.” Comment: Put this sentence as a separate sentence. Doesnt match with first part of the sentence.
Reply: This has not been put as a separate sentence. Instead it was corrected as follows: “as the REE levels in sediments did not correlate with those in the plant.”
- 217 Comment: This paragraph should be in Section 3.3.
Reply: We do not agree that this paragraph should be move to Introduction. The reason is given at the end of p.2 in this Response.
- 219 Comment: higher compared to what?
Reply: changed as: “higher in abundance in above-ground parts than in roots and rhizomes”
L.228-230 Comment: This part should be in a separate sentence.
Reply: We agree. The text has been modified according to the suggestion.
- 233 Comment: what are these 2 different patterns?
Reply: The sentence has been modified as follows: “The REE distributions in plants and in sediments are typically dissimilar,..”
L.233 Comment: This whole paragraph contains unnecessary information to this section. Should be in Introduction.
Reply: We have removed the first sentence from this paragraph. In our opinion, the rest of the paragraph better fits in the Results and Discussion section, as reasoned at the beginning of p.3 in this Response.
L.267-271 Comment: Split in to two sentences.
Reply: This section has been essentially rewritten, and the sentence has been removed.
L.280 “typically considered in mineralogy”?? - Reply: typically considered in mineralogy rather than in life sciences, changed.
- 291-292 Comment: not clear ""as the minimum""
Reply: This sentence has been removed.
L.310-311 “Consider the values reciprocal to TF’ and TFr, namely the ratios of REE contents in rhizomes and leaves of Z. noltei and in the sediment and rhizomes (Figure 4).” Comment: This whole sentence is incomplete and not clear.
Reply: This sentence has been rewritten as follows: “In Figure 4, the ratio of the REE contents in rhizomes and leaves (1/TF’) and the ratio of the REE contents in the sediments and rhizomes (1/TFr) are shown.”
Eq.(5) Comment: what this "C"" denotes?
Reply: C is the content of individual REE, which varies from sediments to leaves and is considered as a function of some variable q that also changes from variables to leaves. This is explained in more detail on p.3.
L.327-328 “which goes down from the pore 327 water to the juice of the plant leaves.”???? – Reply: Corrected: “decreases in the direction from pore water to xylem sap of the plant leaves”
L.330 Comment: Rare earth elements
Reply: Corrected
L.331 “binding to plant matrix carboxyl or phosphate groups close in the binding energy to carbonates” Comment: this phrase is not clear
Reply: This phrase has been modified as follows: “binding to plant matrix carboxyl or phosphate groups nearly as tightly as to carbonates”
L.333-334 Comment: so did you measure Calcium?
Reply: Yes, we measured macroelements, including calcium, and this information has been mentioned in Materials and Methods in the revision. We do not report the measured values for macroelements as this information is not relevant to the subject of the manuscript.
L.339 Comment: substitution for yttrium by which elements La-Nd or Tb?
Reply: All REE are supposed to compete with yttrium for ligands in rhizomes. However, yttrium is bound most specifically there, which fact is reflected in its highest contents in rhizomes. The ratios for La–Nd are most strongly affected due to their relatively high abundances. On the other hand, Tb, being a minor REE, may have as high affinity to the rhizome ligands as Y has, but due to its low abundance it is outcompeted by yttrium.
L.367 Comment: w/w??
Reply: No, w.w. stood for “wet weight”. Now this abbreviation has been spelt in full.
L.368 “Karkinit Bay” Comment: In the Black Sea??
Reply: Yes, Karkinit Bay is in the Black Sea. In the revision this has been specified: “Karkinit Bay in the northern Black Sea”
- Conclusions Comment: Check this number. Should be 4
Reply: Yes, we agree with Reviewer 1. The section number has been changed to 4.
References
Hwang, D.-W., Kim, S.-G., Choi, M., Lee, I.-S., Kim, S.-S., & Choi, H.-G. Monitoring of trace metals in coastal sediments around Korean Peninsula. Mar. Pollut. Bull. 2016, 102, 230–239. https://doi.org/10.1016/j.marpolbul.2015.09.045
Ignat'yeva, O.G.; Romanov, A.S.; Ovsyanyi, E.I.; Kotel'yanets, E.A.; Orekhova, N.A. Physicochemical characteristics of bottom sediments of Kazachya Bay (Black Sea) as indicators of its environmental state. Scientific Notes of V.I. Vernadsky Crimean Federal University. Biology. Chemistry 2005, 18, 43-49 (in Russian).
Kuftarkova, E. A.; Rodionova, N. Yu.; Gubanov, V. I.; Bobko N. I. Hydrochemical characteristics of individual bays of the Sevastopol coast. In B. N. Panov (Ed.) Main results of complex research in the Azov-Black Sea basin and the World Ocean (Jubilee Issue). YugNIRO Publishers, Kerch, Ukraine, 2008, P. 110-117 (in Russian).
Merenkova, S.I.; Malakhova, L.V.; Ivanov, V.E.; Malakhova, T.V.; Bobko, N.I.; Kapranov, S.V. The geochemical features of sedimentation in Sevastopol Bay in the Holocene. Moscow University Geology Bulletin 2023, 78(3), 333-348. https://doi.org/10.3103/S0145875223030122
Ryabushko, V.I., Gureeva, E.V.; Kapranov, S.V.; Bobko, N.I.; Prazukin, A.V.; Nekhoroshev, M.V. Rare earth elements in brown algae of the genus Cystoseira (Phaeophyceae) (Black Sea). Eur. J. Phycol. 2022, 57, 433–445. https://doi.org/10.1080/09670262.2021.2016985
Soloveva O. V.; Tikhonova E. A.; Mironov O. A. The Analysis of Organic Matter Content in the Sea Bottom Sediments of Sevastopol Region (Black Sea) //Physical and Mathematical Modeling of Earth and Environment Processes: Proceedings of 7th International Conference, Moscow, 2021. – Cham : Springer International Publishing, 2022. - P. 107-113.

Reviewer 2 Report
The present manuscript is a study of the distribution of rare earth elements in the marine plant Zostera noltii. The study is simple, it is based only on quantifying these elements in the sediments and in the plant. Indeed, there are few works that address the distribution of these elements. From this point of view the work would be interesting. However, since this paper is only about distribution, words like toxicity should not be in the abstract.
Some considerations to improve the manuscript are:
1) Species name: The most widely used and accepted name is Zostera noltii. Additionally, the first time the species is cited in Materials and Methods should be with the full name, including the author who described the species (this could be avoided in the Title to shorten it).
2) Species names should be written in cursive. Check some in the text.
3) The study is very simple and there is little to comment on the methodology. However, greater detail on sampling is missing. Total number of samples, one? Was it sampled at a single point? Can this be considered representative? How can this area be considered from an environmental point of view, slightly or highly contaminated? Would this influence the results?
4) Figure 2: The abbreviation for logarithm is log.
5) Figure 4: The enormous variation in La is striking. Explanation or discussion would be welcome.
6) The conclusion section is very speculative. Indicate only what is really concluded from this work. For example, lines 376-379, this is not a conclusion of this work, the authors do not directly study these issues. The conclusion section should be more specific.
Author Response
Dear Editor and the Reviewer 2,
We thank the Reviewer 2 for the valuable comments and suggestions aimed to help us improve the quality of the manuscript. We have tried to address all comments and make all the necessary changes.
With best regards,
Prof Sophia Barinova,
Corresponding author
Responses to Reviewer 2 comments
Comment: Species name: The most widely used and accepted name is Zostera noltii. Additionally, the first time the species is cited in Materials and Methods should be with the full name, including the author who described the species (this could be avoided in the Title to shorten it).
Reply: Nanozostera noltei (Hornemann) Tomlinson & Posluszny 2001 is the name currently accepted taxonomically according to AlgaeBase. But we used a more common version of the name Zostera noltei. We absolutely agree with the Reviewer 2 that most of research papers used the name Zostera noltii, which has been indicated in Introduction.
Comment: Species names should be written in cursive. Check some in the text.
Reply: We have checked the manuscript and corrected the species name font where necessary.
Comment: The study is very simple and there is little to comment on the methodology. However, greater detail on sampling is missing. Total number of samples, one? Was it sampled at a single point? Can this be considered representative? How can this area be considered from an environmental point of view, slightly or highly contaminated? Would this influence the results?
Reply: Thanks for the comment. We have added this information below. This sampling can be considered representative within the ecosystem of Kazachya Bay due to location of the sampling site in the center of the bay, relative thermohaline uniformity of its waters and extremely rare events of municipal emergency sewage discharge. As the bay is one of the cleanest in southwestern Crimea, the data obtained can be considered as the weak environmental impact references in case of environmental monitoring using this seagrass species.
“The sampling area was Kazachya Bay, the westernmost bay in Heracles Peninsula in the southwest of Crimea. Kazachya Bay is considered one of the cleanest bays in the area of Sevastopol (Kuftarkova et al., 2008; Soloveva et al., 2021), and this is one of the factors determining the great biodiversity in both the marine and adjacent terrestrial zones. Water temperature in the bay was reported to vary from 7.6 in February to 27.2 °C in July and August. The salinity variations are from 17.43 to 18.12 ‰ (Kuftarkova et al., 2008). Sediments in Kazachya Bay are represented by silted sand with pebbles, silted shell sand and, to a lesser extent, aleuritopelitic sand (Ignat'yeva et al., 2005).
Five live specimens of the seagrass Z. noltei were randomly sampled in Kazachya Bay in June 2021 at one site with the area of about 5 m2 at a depth of 0.5–1 m (Figure 1). The samples were placed in acid-rinsed polyethylene zip slider bags and within one hour delivered to the laboratory on ice. After removal of visible epiphytes, leaves and rhizomes of Z. noltei were rinsed with distilled water to further remove salts, detrital particles and epiphytes. Approximately 20 g of fresh seagrass biomass (with leaves separated from rhizomes using a plastic knife) were cut into small pieces and dried to constant weight at 105 °C. For the REE quantitation, dry biomass with a total weight of 200 mg (100 mg leaves and 100 mg rhizomes) was used.
Sediments with the total mass of approximately 300 g were sampled manually from four different, randomly selected, nearby sites at a depth of 0.5–1 m using a plastic spatula. The thickness of the sampled sediment layer was 2-3 cm. The sediments were thoroughly mixed to eliminate local heterogeneities in the content of elements, dried to constant weight, ground in a porcelain mortar, and sifted through a sieve with a mesh size of 0.5 mm. Five sample replicates were taken for leaves, rhizomes and sediments.”
References
Kuftarkova, E. A.; Rodionova, N. Yu.; Gubanov, V. I.; Bobko N. I. Hydrochemical characteristics of individual bays of the Sevastopol coast. In B. N. Panov (Ed.) Main results of complex research in the Azov-Black Sea basin and the World Ocean (Jubilee Issue). YugNIRO Publishers, Kerch, Ukraine, 2008, P. 110-117 (in Russian).
Soloveva O. V.; Tikhonova E. A.; Mironov O. A. The Analysis of Organic Matter Content in the Sea Bottom Sediments of Sevastopol Region (Black Sea) //Physical and Mathematical Modeling of Earth and Environment Processes: Proceedings of 7th International Conference, Moscow, 2021. – Cham: Springer International Publishing, 2022. - P. 107-113.
Ignat'yeva, O.G.; Romanov, A.S.; Ovsyanyi, E.I.; Kotel'yanets, E.A.; Orekhova, N.A. Physicochemical characteristics of bottom sediments of Kazachya Bay (Black Sea) as indicators of its environmental state. Scientific Notes of V.I. Vernadsky Crimean Federal University. Biology. Chemistry 2005, 18, 43-49 (in Russian).
Comment: Figure 2: The abbreviation for logarithm is log
Reply: Corrected. This was log10.
Comment: Figure 4: The enormous variation in La is striking. Explanation or discussion would be welcome.
Reply: This variation is primarily due to the high variance of La content in rhizomes (Table 1). On the one hand, La is the most abundant element in sediments, and its abundance in rhizomes is expected to be large, too. High La contents in sediments determine high fluxes of this element in the rhizosphere. On the other hand, the most concentrated REE in rhizomes is Y, which is likely selectively bound to certain compounds in rhizomes. As La can compete with Y for binding in rhizomes, and its fluxes to rhizomes are relatively high and possibly nonlinear, this may determine the highest variance of this REE in rhizomes.
Comment: The conclusion section is very speculative. Indicate only what is really concluded from this work. For example, lines 376-379, this is not a conclusion of this work, the authors do not directly study these issues. The conclusion section should be more specific.
Reply: We agree with this comment. Conclusions have been rewritten as follows:
This work presents the first quantitative determination of a complete set of rare earth elements (except for promethium) in the seagrass Z. noltei and in the nearby bottom sediments. The significantly lower REE contents in Z. noltei rhizomes compared to their contents in the sediments have demonstrated the low ability of the seagrass to assimilate these elements, which feature may also be related to specific characteristics of marine sediments in this area. Ratios of the contents of Sc and minor REE, except for Tb, in rhizomes and leaves of Z. noltei and in sediments and rhizomes have turned out to be very close. This fact probably indicates the similarity of mechanisms of the REE accumulation and translocation and is possibly associated with the pH decrease from rhizosphere through rhizomes to leaves. These results can be useful for the further study of REE in the Black Sea ecosystems and to monitor these elements in marine environments.
The mean REE content in Z. noltei leaves that are shed every year and are deposited as beach wracks has been estimated at 0.4 mg·kg–1 wet weight. By utilizing hundreds of thousands of tons of Z. noltei wrack annually cast on the beaches of Crimea, up to 100 tons of REE can be yielded from this biomass. However, environmental safety and cost-effectiveness of this technology requires further comprehensive research.

Reviewer 3 Report
Dear Authors,
The manuscript received for review with the title „Rare earth elements in the seagrass Zostera noltei Hornemann 1832 from the Black Sea coast of Crimea”. Due to the expansion of applications of rare earth elements (REE) in various technological processes, increasing amounts of these metals enter the environment, including the marine one, as pollutants. Very little is known about the bioaccumulation and toxicity of REE in marine organisms. In the present work, we assessed the contents of these metals, including yttrium and scandium, in rhizomes and leaves of the widespread seagrass Zostera noltei and in the nearby sediment from the Black Sea coast. After carefully going through the manuscript, from my point of view it needs some revisions...both in the way of presenting the information and in terms of the rules of technical editing of the journal. The manuscript is quite difficult to read and understand, that's why I warmly recommend the systematic presentation of information and the strict use of research published in the last maximum 3-4 years. I also recommend small changes regarding the rules of technical editing of the journal, but also the English language check. A detailed presentation of the analysis method is also necessary. The presentation of the equipment used in more detail, but also the discussion of the results in coordination with the rest of the research carried out in this area. I saw that there are several studies that highlight the fact that the researched area shows clear signs of pollution with heavy metals, from my point of view it was more important in identifying these factors that they are responsible for this pollution.
How do the water quality parameters influence the concentration of the analyzed metals?
I really appreciate the work done, but from my point of view, a little more work is needed to be accepted for publication of this manuscript.
Congratulations to the authors.
I also recommend small changes regarding the rules of technical editing of the journal, but also the English language check.
Author Response
Dear Editor and the Reviewer 3,
We thank the Reviewer 3 for the valuable comments and suggestions aimed to help us improve the quality of the manuscript. We have tried to address all comments and make all the necessary changes.
With best regards,
Prof Sophia Barinova,
Corresponding author
Responses to Reviewer 3 comments
Comment: The manuscript is quite difficult to read and understand, that's why I warmly recommend the systematic presentation of information and the strict use of research published in the last maximum 3-4 years.
Reply: In the revision, many sections of the manuscript have been reworked or completely rewritten. Hopefully, these modifications, including language check, will contribute to the better comprehensibility of the manuscript. The four subsections of Results and Discussion are relatively independent from each other and we believe they present the information in a systematical enough form. More confusing may be the style of this presentation with the results and discussion intertwined. We deliberately from the very beginning chose this style as the manuscript would have been unreasonably inflated otherwise. The information on REE in seagrasses is extremely scarce; we are aware of only two publications on this matter dated 2014 and 2022. That’s why it is not possible to use only research published in the last 3–4 years in the manuscript.
Comment: I also recommend small changes regarding the rules of technical editing of the journal, but also the English language check.
Reply: In the revision, some technical editing has been performed, including capitalization of nouns in the headings. English language has also been checked.
Comment: A detailed presentation of the analysis method is also necessary. The presentation of the equipment used in more detail, but also the discussion of the results in coordination with the rest of the research carried out in this area.
Reply: We have included a more detailed description of the analysis method in the Materials and Methods section:
“The REE and macroelement concentration in the samples was measured on an ICP-MS instrument PlasmaQuant® MS Elite (Analytik Jena, Germany) in a single scan. The plasma flow was 9 L·min−1, and the RF power was 1.25 kW. The dwell time for the ions was 10 ms, and the scans were recorded in a peak-hopping mode. The absence of polyatomic interferences was ensured by the additional use of collision reaction interface (CRI), in which gaseous hydrogen with the flow rate 40 mL·min−1 was used as a skimmer gas. No internal standard was used. The signal drift was compensated by resloping (measuring the apparent concentration in the standard and applying piecewise linear correction) after every fifth sample.
The REE calibration curves were plotted using the blank solution and standard solutions with the concentrations of 0.001, 0.005, 0.01, 0.05, 0.1 and 0.5 μg·L–1 prepared from multielement standards IV-ICPMS-71A,D (Inorganic Ventures, USA). For the macroelement calibration curves, the standard concentrations of 100, 300 and 500 μg·L–1 were used. In case the analyte concentration in a sample exceeded the upper calibration limit, the sample was diluted with deionized water so as the concentration fell within the calibration range. The coefficient of determination R2 for all calibration curves was no smaller than 0.998.”
As mentioned above, we found only two studies regarding REE concentrations in seagrasses (Sena, 2022; Komar, 2014), although they were not carried out in the Black Sea. The results of these studies have been discussed in appropriate sections.
Comment: I saw that there are several studies that highlight the fact that the researched area shows clear signs of pollution with heavy metals, from my point of view it was more important in identifying these factors that they are responsible for this pollution.
Reply: We agree that the problem of heavy metal pollution in Sevastopol area is pressing. Unfortunately, there has been no systematic spatiotemporal heavy metal monitoring in this area, and therefore, we are unable to identify the factors responsible for this pollution. However, different sites drastically differ in the heavy metal pollution degree even on such a small spatial scale as Sevastopol Bay (7.5 km long) (Kapranov et al., 2023), not to speak of the whole Heracles Peninsula. Kazachya Bay is considered one of the cleanest bays in this area (Kuftarkova et al., 2008: Soloveva et al., 2021), and this is one of the factors determining high biodiversity in both the marine and adjacent terrestrial zones. In view of this, the data obtained in this study can be considered as the small impact references when monitoring REE pollution using this seagrass species.
Comment: How do the water quality parameters influence the concentration of the analyzed metals?
Reply: Sediments are much more indicative of trace element pollution than seawater on a middle- to long-term scale. In sediments, the temporal and spatial variation of trace metal concentrations is far less pronounced than in seawater. Marine plants as pollution indicators are closer to sediments in this respect. Furthermore, the REE contents in marine plants (Cystoseira spp.) demonstrate stronger correlations with those in sediments than in seawater (Ryabushko et al., 2022). Therefore, monitoring trace metal concentrations in sediments provides more meaningful and useful data for evaluating metal contamination in the coastal environment and predicting the effects trace metals may have on marine ecosystem (Hwang, 2016).
References
Hwang, D.-W., Kim, S.-G., Choi, M., Lee, I.-S., Kim, S.-S., & Choi, H.-G. Monitoring of trace metals in coastal sediments around Korean Peninsula. Mar. Pollut. Bull. 2016, 102, 230–239. https://doi.org/10.1016/j.marpolbul.2015.09.045
Kapranov, S.V.; Kozintsev, A.F.; Bobko, N.I.; Ryabushko, V.I. Elements in Soft Tissues of the Young Mediterranean Mussel Mytilus galloprovincialis Lam. 1819 Collected in Sevastopol Bay (Crimea, Black Sea): Effects of Age, Sex, Location, and Principal Morphometric Parameters. Animals 2023, 13, 1950. https://doi.org/10.3390/ani13121950
Kuftarkova, E. A.; Rodionova, N. Yu.; Gubanov, V. I.; Bobko N. I. Hydrochemical characteristics of individual bays of the Sevastopol coast. In B. N. Panov (Ed.) Main results of complex research in the Azov-Black Sea basin and the World Ocean (Jubilee Issue). YugNIRO Publishers, Kerch, Ukraine, 2008, P. 110-117 (in Russian).
Ryabushko, V.I., Gureeva, E.V.; Kapranov, S.V.; Bobko, N.I.; Prazukin, A.V.; Nekhoroshev, M.V. Rare earth elements in brown algae of the genus Cystoseira (Phaeophyceae) (Black Sea). Eur. J. Phycol. 2022, 57, 433–445. https://doi.org/10.1080/09670262.2021.2016985
Soloveva O. V.; Tikhonova E. A.; Mironov O. A. The Analysis of Organic Matter Content in the Sea Bottom Sediments of Sevastopol Region (Black Sea) //Physical and Mathematical Modeling of Earth and Environment Processes: Proceedings of 7th International Conference, Moscow, 2021. – Cham : Springer International Publishing, 2022. - P. 107-113.

Reviewer 4 Report
Dear Authors!
I had enjoyed reading your study! It looks like good The analytical methods and statistical processing of the data performed on the high level, in my opinion.
The main issue of the manuscript is discrepancy of goals and tasks with content and conclusions. In my opinion, the discussion section could be expanded by adding the comparative analysis of literature about translocation of REE in plants or algae. The several paragraphs from the Discussion could be moved to the Introduction. The abstract and the conclusions should be rewritten according to the findings of the study. The additional description of selected approaches in sampling, calculation of ratios and comparative data is needed.
Here my minor comments and remarks:
Line 9-11: the first sentence is too high and looks like introduction part. Perhaps, “enter the environment” should be changed to “affect” if you’ve written about pollution.
Line 11: “very little is known” – needs rewriting. Why it is important to note and why it is important to know? How it was connected with the subject?
Line 13. Are you sure that “sediment” should be here in singular form?
Line 20. “the greatest” you meant that it was in the world or in the ages? “the highest” is probably fits better.
Line 21. “minor heavy elements” Which classification did you use here?
Line 22. “sediments” in plural
Line 25-26. The sentence should be rewritten. “deposits allow it to be considered as a source of REE”. So the main aim of the study is to define whether Zostera could be possible source of REE, is it true?
Line 82. “archivers of anthropogenic activity” – in my opinion, it will be better to use more precise words. In this section you mentioned that “they can be used as an indicator (indicators?) of short term changes (how?)…”. In the next sentence, “There is currently very little information on interactions between seagrasses and pollutants”. So how this could be?
Line 88. The purpose is "to study the distribution …. as a possible indicator and a potential agent for the REE recovery and bioremediation” the distribution as an indicator? This sentence needs to be separated into several.
Line 95. How many samples of sediments did you collect? How much weight in general did you collect before analysis? The samples were collected manually? Parameters of substrate? The plants were live or not? How did they store before analysis? These questions should be described in the section.
Line 130. Why did you choose the values in UCC from the Taylor, 1985? Why didn't Rudnick et al. 2003? As you mentioned, due to “average source rock composition”. However, how it fits to your studied zone? The local sediments could contain higher levels of REE due to specific rock composition.
Line 133. I cannot find this statement in the Zhang 2002 [37]. Moreover, they state: “Enrichment factor (EF) of individual metals is highly variable depending upon the element of interest and river of study”. In my opinion, the EF could be considered based on the local rock composition.
Line 163. It is not true. Sc and La demonstrated significantly different levels according to Table 1. The sentence should be checked.
Line 196. How was it connected with the other sections and study subject? This section should be inserted in Introduction.
Line 236. Again, this section did not connect with the other text. It will look better to add it to the Introduction.
Line 243. Ce3+ and La3+ are ions of elements.
Line 259. Where could the reader find the “geochemical limits”? What did you exactly mean?
Line 269-270. The sentence is too long.
Line 287. How the UCC values were normalized?
Line 299-300. Figure 3 should be inserted close to describing text. The caption should be checked: “EF of REE with respect to UCC”. But the EF – is a coefficient, included the values from the UCC!
Line 320. The detailed description of the equation (q and k) is needed.
Line 323. Figure 4. The whiskers should be described in the caption
Line 328. It is not well known for all readers. The reference is needed.
Line 376. Conclusions. The two sentences are not conclusions according to your study.
Line 381. “the lower concentrations in Zostera demonstrate the low ability of seagrass to accumulate elements”. It is not true. Could the REE be intake from the water and not from sediments? Your conclusion stated about all species of seagrass. Is it true for all? How this connected with the abilities of Zostera? How did you measure the ability to accumulation? What about self-purification abilities? The sentence should be rewritten.
Line 384-388. The sentence is too long, it will be better to divide it into several statements. How did you measure the pH (“local” and in “pore waters”)? Where did you demonstrate this influence?
I am not native English speaker, however, in my opinion, the manuscript was written on the good scientific understandable language except several places with long sentences.
Author Response
Dear Editor and the Reviewer 4,
We are very thankful to the Reviewer 4 for the valuable comments and suggestions, which have been extremely useful for improving the manuscript.
With best regards,
Prof Sophia Barinova,
Corresponding author
Responses to Reviewer 4 comments
Comment: Line 9-11: the first sentence is too high and looks like introduction part. Perhaps, “enter the environment” should be changed to “affect” if you’ve written about pollution. Line 11: “very little is known” – needs rewriting. Why it is important to note and why it is important to know? How it was connected with the subject?
Reply: We have removed these introductory phrases.
Comment: Line 13. Are you sure that “sediment” should be here in singular form?
Reply: We thank the Reviewer for drawing or attention to this inconsistency. “Sediment” has been substituted for “sediments” in this line and throughout where appropriate.
Comment: Line 20. “the greatest” you meant that it was in the world or in the ages? “the highest” is probably fits better.
Reply: We agree with the suggestion. Corrected.
Comment: Line 21. “minor heavy elements” Which classification did you use here?
Reply: There is no generally accepted agreement on the definition of light and heavy rare earth element (REE) classification. Conventionally, lanthanum, cerium, praseodymium, neodymium, promethium and samarium are assigned to light REE. Heavy REE include europium, gadolinium, terbium, dysprosium, holmium, erbium, thulium, ytterbium, lutetium, scandium and yttrium (Omodara et al., 2019; Patah et al., 2021). Because our division differed in some aspects from the above-mentioned one (e.g., samarium behaved as heavy REE while terbium did not), we have decided to refer to these elements here only as “minor REE” in the revised ms.
References
Omodara, L.; Pitkäaho, S.; Turpeinen, E.-M.; Saavalainen, P.; Oravisjärvi, K.; Keiski, R.L. 2019. Recycling and substitution of light rare earth elements, cerium, lanthanum, neodymium, and praseodymium from end-of-life applications - A review. Journal of Cleaner Production 236, 117573. https://doi.org/10.1016/j.jclepro.2019.07.048
Patah, M.F.A.; Shafiee, N.S.; Ismail, R.; Bahar, A.M.A.; Khan, M.M.A.; Rak, A.Eh.; Awang, M. 2021. Distribution of light (LHREE) and heavy rare earth elements (HREE) in Kelantan granitoids rock. 3rd International Conference on Tropical Resources and Sustainable Sciences. IOP Conf. Series: Earth and Environmental Science 842, 012038. doi:10.1088/1755-1315/842/1/012038
Rare earth elements: A briefing note by the Geology Society of London. Available online: https://www.geolsoc.org.uk/~/media/shared/documents/policy/Rare%20Earth%20Elements%20briefing%20note%20final%20%20%20new%20format.pdf (accessed on 03.10.2023).
Comment: Line 22. “sediments” in plural
Reply: Corrected according to the suggestion
Comment: Line 25-26. The sentence should be rewritten. “deposits allow it to be considered as a source of REE”. So the main aim of the study is to define whether Zostera could be possible source of REE, is it true?
Reply: The main objectives of the study were to estimate the REE content in the seagrass, to assess its possible REE pollution bioindicator abilities and consider it as a possible source of REE. The sentence has been rewritten in accordance to the Reviewer’s suggestion.
Comment: Line 82. “archivers of anthropogenic activity” – in my opinion, it will be better to use more precise words. In this section you mentioned that “they can be used as an indicator (indicators?) of short term changes (how?)…”. In the next sentence, “There is currently very little information on interactions between seagrasses and pollutants”. So how this could be?
Reply: Leaves of the seagrass Zostera noltei are shed every year, therefore, they can serve as an indicator of the relatively short-term REE pollution in the marine environment. At the same time, roots and rhizomes of Z. noltei can be called “archivers of anthropogenic activity” because of their perennial and attached life in marine sediments. This has been added in the revision. To date, we are aware of only two studies reporting the REE contents in the seagrasses Halodule wrightii (Sena et al., 2022) and Cymodocea nodosa (Komar et al., 2014), while mechanisms of the REE inclusion in seagrasses and REE impact on them are unknown. Corrected at the end of Introduction.
Comment: Line 88. The purpose is "to study the distribution …. as a possible indicator and a potential agent for the REE recovery and bioremediation” the distribution as an indicator? This sentence needs to be separated into several.
Reply: The seagrass itself rather than element distribution in it is meant as the indicator. We have rephrased this sentence as follows: “The purpose of this work is to study the distribution of REE in rhizomes and leaves of the seagrass Zostera noltei from the coastal area of the Black Sea. This information allows assessing the usability of the seagrass as a possible indicator of REE pollution and as a potential agent for the REE recovery and bioremediation.” Corrected at the end of Introduction with dividing into two sentences.
Comment: Line 95. How many samples of sediments did you collect? How much weight in general did you collect before analysis? The samples were collected manually? Parameters of substrate? The plants were live or not? How did they store before analysis? These questions should be described in the section.
Reply: We have added these necessary data as follows:
The sampling area was Kazachya Bay, the westernmost bay in Heracles Peninsula in the southwest of Crimea. Kazachya Bay is considered one of the cleanest bays in the area of Sevastopol (Kuftarkova et al., 2008; Soloveva et al., 2021), and this is one of the factors determining the great biodiversity in both the marine and adjacent terrestrial zones. Sediments in Kazachya Bay are represented by silted sand with pebbles, silted shell sand and, to a lesser extent, aleuritopelitic sand (Ignat'yeva et al., 2005).
Five live specimens of the seagrass Z. noltei were randomly sampled in Kazachya Bay (Black Sea coast of Crimea) in June 2021 at one site with the area of about 5 m2 at a depth of 0.5–1 m (Figure 1). The samples were placed in acid-rinsed polyethylene zip slider bags and within one hour delivered to the laboratory on ice. After removal of visible epiphytes, leaves and rhizomes of Z. noltei were rinsed with distilled water to further remove salts, detrital particles and epiphytes. Approximately 20 g of fresh seagrass biomass (with leaves separated from rhizomes using a plastic knife) were cut into small pieces and dried to constant weight at 105 °C. For the REE quantitation, dry biomass with a total weight of 200 mg (100 mg leaves and 100 mg rhizomes) was used.
Sediments with the total mass of approximately 300 g were sampled manually from four different, randomly selected, nearby sites using a plastic spatula. The thickness of the sampled sediment layer was 2-3 cm. The sediments were thoroughly mixed to eliminate local heterogeneities in the content of elements, dried to constant weight, ground in a porcelain mortar, and sifted through a sieve with a mesh size of 0.5 mm. Five replicates were taken for leaves, rhizomes and sediments.
References
Ignat'yeva, O.G.; Romanov, A.S.; Ovsyanyi, E.I.; Kotel'yanets, E.A.; Orekhova, N.A. Physicochemical characteristics of bottom sediments of Kazachya Bay (Black Sea) as indicators of its environmental state. Scientific Notes of V.I. Vernadsky Crimean Federal University. Biology. Chemistry 2005, 18, 43-49 (in Russian).
Kuftarkova, E. A.; Rodionova, N. Yu.; Gubanov, V. I.; Bobko N. I. Hydrochemical characteristics of individual bays of the Sevastopol coast. In B. N. Panov (Ed.) Main results of complex research in the Azov-Black Sea basin and the World Ocean (Jubilee Issue). YugNIRO Publishers, Kerch, Ukraine, 2008, P. 110-117 (in Russian).
Soloveva O. V.; Tikhonova E. A.; Mironov O. A. The Analysis of Organic Matter Content in the Sea Bottom Sediments of Sevastopol Region (Black Sea) //Physical and Mathematical Modeling of Earth and Environment Processes: Proceedings of 7th International Conference, Moscow, 2021. – Cham : Springer International Publishing, 2022. - P. 107-113.
Comment: Line 130. Why did you choose the values in UCC from the Taylor, 1985? Why didn't Rudnick et al. 2003? As you mentioned, due to “average source rock composition”. However, how it fits to your studied zone? The local sediments could contain higher levels of REE due to specific rock composition.
Reply: We thank the Reviewer for this insightful comment. The values from Rudnick and Gao (2003, 2014) are indeed more up-to-date than those of Taylor and McLennan (1985), although not too much different from them. Unfortunately, the data on the average rock composition at this particular site are not available to us from the literature. Instead, in the revision, we have used the average data on the early (Late Pleistocene) sediments from a core at a nearby site (Sevastopol Bay) (Merenkova et al., 2023). They indeed differed from those given by Taylor and McLennan (1985) and Rudnick and Gao (2014). The enrichment factors for most of REE have turned out to be much lower and no signs of the “inverse Oddo-Harkins pattern” have been detected. The respective changes have been made in the discussion of these results.
References
Merenkova, S.I.; Malakhova, L.V.; Ivanov, V.E.; Malakhova, T.V.; Bobko, N.I.; Kapranov, S.V. The geochemical features of sedimentation in Sevastopol Bay in the Holocene. Moscow University Geology Bulletin 2023, 78(3), 333-348. https://doi.org/10.3103/S0145875223030122
Comment: Line 133. I cannot find this statement in the Zhang 2002 [37]. Moreover, they state: “Enrichment factor (EF) of individual metals is highly variable depending upon the element of interest and river of study”. In my opinion, the EF could be considered based on the local rock composition.
Reply: Indeed, Zhang and Liu (2002) wrote: “the concentration of Al in weathering products and their parent materials are generally comparable.” However, the Al contents in our sediments and deposits from the Late Pleistocene (Merenkova et al., 2023), differed by a factor of 42 (301 vs. 12800 ppm, respectively). For this reason, we have discarded using Al content as the normalization index. Furthermore, there can be a different percentage of weathering products in sediments due to various biotic and abiotic factors, and for the normalization, we sought macroelements that would have nearly constant abundance throughout the Holocene and approximately the same content as in the sediments under research. Such elements selected were Mg and P, and their contents have been used for the normalization in the revision. These normalizations yield slightly different EF profiles, but they are consistent in terms of the overall enrichment or depletion (Sc, Y, La, Tb are enriched, Ce–Dy are depleted, Ho–Lu are either depleted or at the primordial levels). The respective corrections have been made in sections 2 (Materials and Methods) and 3.2 (REE Enrichment and Anomalies in Sediments).
Comment: Line 163. It is not true. Sc and La demonstrated significantly different levels according to Table 1. The sentence should be checked.
Reply: Thank you for this remark. We have removed this sentence.
Comment: Line 196. How was it connected with the other sections and study subject? This section should be inserted in Introduction.
Reply: We agree. It has been moved to Introduction.
Comment: Line 236. Again, this section did not connect with the other text. It will look better to add it to the Introduction.
Reply: We have removed this sentence from the revision.
Comment: Line 243. Ce3+ and La3+ are ions of elements.
Reply: Corrected
Comment: Line 259. Where could the reader find the “geochemical limits”? What did you exactly mean?
Reply: This part of the manuscript has been removed in the revision. Actually, that was an infelicitous wording, and we meant the levels in the Upper Continental Crust.
Comment: Line 269-270. The sentence is too long.
Reply: In the revision, this sentence has been shortened.
Comment: Line 287. How the UCC values were normalized?
Reply: We meant normalization of the element content in sediments to these values. In the revision, we have abandoned using average UCC values as normalization factors, as mentioned above. Instead, Late Pleistocene values from the same area have been used.
Comment: Line 299-300. Figure 3 should be inserted close to describing text. The caption should be checked: “EF of REE with respect to UCC”. But the EF – is a coefficient, included the values from the UCC!
Reply: The figure position has been changed. The caption has been modified (shortened) as all needed clarifications are in the text.
Comment: Line 320. The detailed description of the equation (q and k) is needed.
Reply: We have added a description of k (proportionality factor that shows the steepness of the concentration increment with the parameter q increase) immediately after Eq. (4). Before Eq. (4), q has been mentioned as a yet unspecified variable. After Eq. (6) and Figure 4, it was mentioned that q is likely the variable that denotes the difference between pH in different parts of the plant (rhizomes, leaves) and pH of pore water in the sediments.
Comment: Line 323. Figure 4. The whiskers should be described in the caption
Reply: They have been described in the caption as the 95% confidence intervals.
Comment: Line 328. It is not well known for all readers. The reference is needed.
Reply: The references have been added.
Comment: Line 376. Conclusions. The two sentences are not conclusions according to your study.
Reply: We agree. Conclusions have been rewritten as follows:
This work presents the first quantitative determination of a complete set of rare earth elements (except for promethium) in the seagrass Z. noltei and in the nearby bottom sediments. The significantly lower REE contents in Z. noltei rhizomes compared to their contents in the sediments have demonstrated the low ability of the seagrass to accumulate these elements, which feature may also be related to specific characteristics of marine sediments in this area. Ratios of the contents of Sc and minor REE, except for Tb, in rhizomes and leaves of Z. noltei and in sediments and rhizomes have turned out to be very close. This fact probably indicates the similarity of mechanisms of the REE accumulation and translocation and is possibly associated with the difference between the local pH value and the pH of pore waters. These results can be useful for the further study of REE in the Black Sea ecosystems and to monitor these elements in marine environments. The mean REE content in Z. noltei leaves that are shed every year and are deposited as beach wracks has been estimated at 0.4 mg·kg–1 wet weight. By utilizing hundreds of thousands of tons of Z. noltei wrack annually cast on the beaches of Crimea, up to 100 tons of REE can be yielded from this biomass. However, environmental safety and cost-effectiveness of this technology requires further comprehensive research.
Comment: Line 381. “the lower concentrations in Zostera demonstrate the low ability of seagrass to accumulate elements”. It is not true. Could the REE be intake from the water and not from sediments? Your conclusion stated about all species of seagrass. Is it true for all? How this connected with the abilities of Zostera? How did you measure the ability to accumulation? What about self-purification abilities? The sentence should be rewritten.
Reply: We do not insist on the totality of REE in seagrass shoots being accumulated from sediments (moreover, there are some indications that many elements in the above-ground parts of marine plants are absorbed from water). We have specified in the revised Conclusions that this observation concerns only rhizomes (which almost definitely accumulate elements buried in sediments, where the effects of seawater composition are minor). We do not either claim that these findings apply to all seagrass species. We found only two studies, besides ours, on REE levels in seagrasses. These studies indicated that increased concentrations in sediments did not cause increased accumulation of elements in seagrasses (Sena, 2022; Komar, 2014). Reorganized and rewritten in the revised ms.
Comment: Line 384-388. The sentence is too long, it will be better to divide it into several statements. How did you measure the pH (“local” and in “pore waters”)? Where did you demonstrate this influence?
Reply: In the revised Conclusions, the sentence from L.384-388 has been divided into two ones. By “local” pH, we mean pH in specific parts of the plant, e.g. roots, rhizomes and leaves. We did not measure pH in plant tissues or in pore water, but it is known (Brodersen et al., 2018) that pH decreases in the direction from pore water, which is typically no less alkaline than seawater (in the area under study, seawater pH varies in the range of 8.1–8.4), to xylem sap, which is typically acidic (pH 4.5–7.4) (Pramsohler et al., 2022). The more acidic conditions, the more REE ions are solubilized and the less REE ions are retained in aggregates or organic complexes in plant tissues. We do not demonstrate this influence explicitly but rather assume that it is this parameter (pH) that is associated with the variable q when discussing the pairwise equality of the REE content ratios in Figure 4.
References
Brodersen, K.E.; Kühl, M.; Nielsen, D.A.; Pedersen, O.; Larkum, A.W.D. Rhizome, root/sediment interactions, aerenchyma and internal pressure changes in seagrasses. In Seagrasses of Australia, A. W. D. Larkum et al. (Eds.), Springer, Cham, Switzerland, 2018, pp. 393-418. https://doi.org/10.1007/978-3-319-71354-0_13
Pramsohler, M.; Lichtenberger, E.; Neuner, G. Seasonal xylem sap acidification is governed by tree phenology, temperature and elevation of growing site. Plants 2022, 11(15), 2058. doi: 10.3390/plants11152058

Round 2
Reviewer 1 Report
Authors have addressed the comments adequately.
Mostly ok. But please double check.
Reviewer 2 Report
The changes introduced in the manuscript have been correct. The quality of the manuscript has been improved and responses to comments have been appropriate. Therefore, the manuscript can be accepted.
Reviewer 3 Report
Dear Authors,
Recommend publication of the manuscript in this form.
Congratulations to the authors.